# Educating at Scale for Sustainable Development and Social Enterprise Growth: The Impact of Online Learning and a Massive Open Online Course (MOOC)

**Sara Calvo [1,\*], Fergus Lyon [2], Andrés Morales [1] and Jeremy Wade [3]** 

[1]  Business Department, Universidad Internacional de La Rioja, Avenida de la Paz, 137, 26004 Logroño, Spain; andres.morales@unir.net

[2]  Centre for Enterprise and Economic Development Research, Middlesex University Business School, The Burroughs, London NW4 4BT, UK; f.lyon@mdx.ac.uk

[3]  Jindal Centre for Social Innovation & Entrepreneurship, O.P. Jindal Global University, Delhi 131001, India; jwade@jgu.edu.in

\*  Correspondence: sara.calvo@unir.net

**Abstract:** The solutions to the grand challenges of sustainability, poverty, and health affecting the world will require education and capacity building for all individuals implementing change on a global scale. The challenge ahead is to reach those missed by traditional education and support networks. Online Learning and Massive Open Online Courses (MOOCs) have the potential to build knowledge and share best practice experiences among individuals worldwide. This paper examines the case of the FutureLearn Social Enterprise Program, a series of MOOCs with over 50,000 registered learners, of which 15% become active learners, engaging in online exercises, debates, and conversations. This paper draws on quantitative and qualitative data collected over four years. The findings show that the course has not only had an impact on the creation of new startup social enterprises, but it has also supported a large proportion of learners in developing sustainability and social entrepreneurial ideas within a range of organisations in the public, private, and civil society sectors. The findings also show a positive cascading impact effect from the learners registered on the course to those in their network, as ideas are shared, and learners become mentors to others. Our conclusions demonstrate how digital education and online courses contribute to global education for sustainable development and social enterprise development.

**Keywords:** education for sustainable development; MOOCs; Social Enterprise; social entrepreneurs; impact evaluation

## 1. Introduction

Grand challenges of our times related to environmental and social change require rapid change, the development of alternatives to 'business as usual' and education for sustainable development [1]. Social Enterprises (SEs) can provide this alternative by bringing new business models for sustainability that combine a core environmental or social objective with a trading activity [2]. This paper examines how digital education and online courses can inspire social entrepreneurship and educate on a global scale. It contributes to the understanding of how education and support can lead to social entrepreneurship growth and the scaling up of sustainable solutions to global challenges.

Universities have been impacted by the development of information technology and the introduction of innovative learning tools to deliver effective, just-in-time, and personalized learning

processes [3]. Massive Open Online Courses (MOOCs) are changing how people can access digital knowledge, creating new opportunities for learning and competence development.

This paper explores how a university-based MOOC has provided the impetus for widespread education on social enterprise and sustainability. This is shown to lead to the development and growth of new and existing social enterprises as well as having cascading impacts derived from the course participants sharing their new knowledge about social enterprise and sustainability. The MOOC program developed by Middlesex University Business School in collaboration with Minca Ventures Ltd and the Jindal Centre for Social Innovation and Entrepreneurship (JSiE) consisted of three online courses designed to educate learners how to start and scale social enterprises [4]. The content was derived from academic research on social enterprise by the team developing the MOOC combined with interactive elements for the learners.

Research on sustainable development-related MOOCs, and, in particular, social enterprise courses, has been limited with very little analysis of the impact on actual practice. In this paper, the authors, therefore, seek to answer three related questions: How can a MOOC encourage social enterprises to start up? How does a MOOC contribute to the impact stakeholders and learners had on their existing projects and organisations? How have learners used the MOOC for sharing knowledge on sustainable development and social enterprise? The paper starts by examining the literature on education for sustainable development, social enterprise and MOOCs. Building on this, the authors introduce the Freeth/Kirkpatrick evaluation model to explore the impact of entrepreneurship education courses and introduce a model of social entrepreneurial growth. The authors then discuss the methodology used for a rigorous assessment of impact before presenting the research findings and discussion. The paper concludes with the consideration of these results, making suggestions for further research.

## 2. Literature Review

### 2.1. Education for Sustainable Development and Social Enterprise

For the past 30 years, high-quality education has been considered crucial to creating a sustainable future for our society [5]. The term ´sustainability´ has been used to refer to the ability to meet the needs of the present without putting in risk the resources for the future generations. Education for sustainability development has been internationally recognized as a key enabler for the realization of sustainable development. Policymakers and governments have considerably advanced the efforts to promote education and sustainability awareness worldwide due to the current structural changes to economies and pressures of globalization. Through raising awareness, learners can respond to sustainability challenges, such as economic disparity, global warming, and social exclusion [4,6–9]. Although education for sustainable development is crucial, the concept of sustainability is loosely defined, which creates challenges for education and approaches for environmental awareness-raising [10].

There are three pillars that need to be considered for considering sustainability: economic, environmental, and social. Social Enterprises target the three sustainability pillars by having an environmental or social mission and reinvesting generated profits to achieve multiple bottom lines. In recent years, research has reported the rapid development of the Social Enterprise (SE) sector on an international level [4,11–13].

This paper draws on theories of social entrepreneurial growth and scaling in order to understand how the online courses have an impact. Studies of social enterprise have shown that growth can be through the start-up of new social enterprises and the organic expansion of an organization. Expansion can be found in terms of increasing the number of service users, increasing the amount of services being provided to specific users, increasing the geographical reach of services, and also diversifying into new service areas [14]. Lyon and Fernandez [15] show how the scaling of social enterprises can come about through sharing ideas outside of the organization and encouraging successful replication approaches that have a social and environmental impact. In this way, social enterprises differ from

the private sector, as the emphasis on environmental/social objectives means that they can judge their success by changes in society, not just by the growth of their organization's turnover or profit.

However, there is little research on how education for social enterprise shapes sustainable development, despite the significant investment in this area. In recent years, universities have been a key actor in promoting and implementing sustainability, as they make significant social, economic, academic, scientific, and technological contributions to their local and national environments [9]. Many scholars see the impact of universities on sustainability as they educate the next generation of decision makers, influencers, and leaders. There has been an increase in the number of universities that have responded to this with the implementation of education for sustainable development.

One trend that has developed recently is the increase in the number of universities launching educational courses related to growing social enterprises. Several studies have suggested that social enterprises developed in universities stimulate and sustain diversity, social inclusion, citizenship, and local learning communities and partnerships [16–19]. A recent study, conducted by McNally et al. [20], reviewed social enterprise courses from universities from around the world and demonstrated that educators influenced the attitudes of learners toward their courses before classes even begin. The study findings highlighted the move from instructor-oriented to more learner-centered teaching philosophies. However, as mentioned earlier, few studies have explored the higher level (long-term) impact of online courses providing capacity building for social enterprise development.

## 2.2. Sustainability-Related MOOCs

With over 50% of the global population now online, the digital world is modifying the way people are learning and retrieving information [21]. MOOCs allow individual learners the ability to self-regulate their learning, determining when and how they engage. MOOCs allow us to have large numbers of individuals learning together and building new relationships between themselves. With MOOCs, the teaching method is moving from the traditional transfer learning model to a flipped classroom model where the learner interacts with other students, peers, and has the flexibility to access all areas of information and resources. With the huge amount of online educational material, this has also become a useful and beneficial method in teaching [22].

MOOCs have captured the interest of some academics and students in higher education but, more importantly, have effectively targeted students and teachers not in traditional higher education. Here, they demonstrated an ability to provide quality education and promote lifelong learning opportunities for all. The Education 2030 Framework for Action, adopted at Incheon (Republic of Korea) in May 2015, recognizes lifelong learning for all as one of the underpinning principles of this new vision, stating that "all age groups, including adults, should have opportunities to learn and continue learning." However, there have been some criticisms of MOOCs. A study conducted by Margaryan et al. in 2015 [23] highlighted that the quality of MOOCs in terms of learning design is low. As they argued, MOOCs emphasised a static design and a passive approach to the acquisition of knowledge. Another study conducted by Rabin et al. in 2019 [24] pointed out that MOOCs count with high drop-out rates, therefore making it difficult to monitor learners.

The increase in the number of MOOCs has allowed many sustainability educators to change their practice developing content related to energy and resources, ethics, ecology, and management [25]. This finding is consistent with those of previous studies conducted by Matten and Moon [26] and Wu et al. [27] that suggest that ethics is one of the most important contents in sustainability education.

A study conducted by Beltrán et al. [28] exploring energy sustainability MOOCs suggests that the impact of MOOCs is greater if designers include entrepreneurship issues in the content and encourage activities that promote networking among participants. Although these studies indicate the relevance of sustainability-related MOOCs, there has been very little research on exploring the impact of MOOCs on learners' knowledge of sustainability and their ability to turn this knowledge into action.

*2.3. Towards A Model for the Evaluation of the Impact of MOOCs on Sustainability and Social Entrepreneurship*

As this paper is examining the impact of education on sustainability and social enterprise, there are insights from the evaluation of other entrepreneurship education courses. The few studies evaluating the impact of entrepreneurial educational courses focus on short-term impacts [29–32] rather than long-term change.

A useful tool for evaluating the impact of educational programs suggested by Fayolle [33] is the Freeth/Kirkpatrick model (FKM), developed in 1959 and updated in 2002. This model distinguishes between four levels of evaluation: Reaction (Level 1), Learning (Level 2), Behaviour (Level 3), and Results (Level 4). As shown in Figure 1, this model captures educational outcomes from individuals' reaction (Level 1) to the impacts of the program on the organization (Level 4) (see Figure 1).

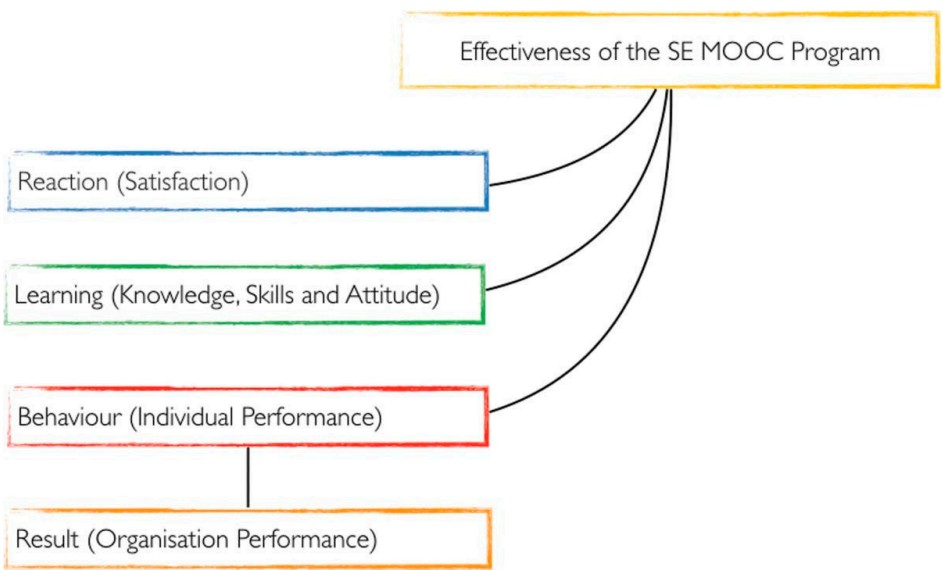

**Figure 1.** Using the Freeth/Kirkpatrick model to understand the SE MOOC program. Source: Calvo et al. (2018).

While the first level (Reaction) considers how participants reacted to the training or intervention they have received, the second level (Learning) focuses on the knowledge and skills acquired. Level 3 (Behaviour) examines the change in behaviour of participants for example in their workplace after the completion of the training or intervention. Finally, Level 4 (Result) considers the long-term outcomes of the intervention or training. This level considers an organizational-level evaluation looking at results of the interventions, and of subsequent reinforcement. In this paper we have used the Freeth/Kirkpatrick model to understand the higher level impact of the SE MOOC program, in particular we explore two dimensions- Levels 3 (Behaviour) and 4 (Results).

This model can be elaborated by drawing on entrepreneurship education theory. Therefore, the authors draw on Nabi et al's [34] review of entrepreneurship education which called for research on longer-term higher level impact associated with entrepreneurial intention, actual start-ups and longer-term impacts on business. Entrepreneurship is often conceived in simplistic forms as the creation of enterprise. However, a large body of literature has shown how entrepreneurship is also a process related to innovation, creativity and organisational growth. This paper therefore explores general awareness raising about social enterprise, the impact of courses on intentions to start up, the actual creation of enterprises, the growth of existing enterprises and the development of social entrepreneurship ideas within existing private and public sector enterprises (what can be termed intrapreneurship).

The impact of education on scaling and growth of social enterprise for sustainable development requires a nuanced view of expansion. This needs to go beyond the narrow view of economic growth of an organisation judged only by financial measures [14]. In line with social entrepreneurship theory, there is a need to understand how scaling up is about environmental and social change both within organisations and though encouraging others to start social enterprise. Building on Lyon and Fernandez [15], this paper also examines the scaling of social enterprise through the sharing of ideas and replication of successful models that have a positive sustainability impact.

Finally, this paper also assesses the additionality of the educational support provided by the online course. Building on previous research on the impact of entrepreneurship support, it is necessary to examine the attribution and additional impact coming from an intervention [35], in addition to what might have happened anyway. By adding this to the evaluation model, it is possible to draw conclusions about the extent to which benefits can be attributed to a particular enterprise support or educational course.

## 3. Methodology

### 3.1. Overview of the Middlesex/Minca Ventures/Jindal Social Enterprise MOOC

In 2016, a team of educators from the Middlesex University Business School, the Jindal Centre for Social Innovation and Entrepreneurship (JSiE) and Minca Ventures Ltd developed a massive social enterprise MOOC program with the purpose of inspiring people to learn about social enterprise and sustainability, starting a social enterprise, and/or growing their existing social enterprise projects. The Social Enterprise MOOC program consisted of three online courses. The first course is entitled 'Social Enterprise: Business Doing Good', while the second course is 'Social Enterprise: Turning Ideas into Action'. The third course is named 'Social Enterprise: Growing a Sustainable Business'. The first course focused on understanding the concept of social enterprise and the second and third courses were focused on developing a social enterprise project or growing an existing social enterprise, respectively. While, for the first course, the target audience was anyone interested in learning about social enterprise, for the second and third courses, the audience was people interested in setting up and scaling social enterprises. Each of the courses were divided into three learning weeks (4 hours per week with a total of 12 hours). Each week contained a set of activities, which in turn contained a set of steps with a set of learning activities. The steps consisted of discussions, social-media-related activities, video case studies, articles, quizzes, practical exercises, peer review and problem-based activities. This demonstrated good practice entrepreneurship education by building competence through solving problems in real life situations [34], having the opportunity to pitch ideas to an international video pitch competition. Table 1 shows the content that was covered in the three courses with details of what was taught in each of the three learning weeks.

**Table 1.** Content of the social enterprise Massive Open Online Course (MOOC) program.

| SE Program | Week 1 | Week 2 | Week 3 |
|---|---|---|---|
| **Course 1 Social Enterprise: Business Doing Good** | **Addressing the world's most pressing challenges** <br> - Global and Local challenges around the world <br> - Local solutions around the world and in your local area <br> - Social Enterprises in the Global North and South <br> - The Sustainable Development Goals | **The Diverse World of Social Enterprise** <br> - Defining Social Enterprises <br> - Types of Social Enterprises <br> - Social Enterprise Discourses <br> - Social Enterprises as Hybrid Organisations <br> - Understanding Social Impact | **Social Entrepreneurs, Network and Ecosystems** <br> - Changemakers and personal transformation <br> - The Social Enterprise ecosystem <br> - The role of government, universities and celebrities <br> - Enabling organisations for social enterprise <br> - Social Enterprise business ideas <br> - Contributors and Additional resources |

**Table 1.** *Cont.*

| SE Program | Week 1 | Week 2 | Week 3 |
|---|---|---|---|
| **Course 2** **Social Enterprise: Turning Ideas into action** | **Your Social Enterprise Idea** - Link between problem and idea - Ideation process - Sharing your idea with key stakeholders - Digital technologies and Social Enterprise - Innovation in Social Enterprises | **Designing your Social Enterprise Business Model** - Business Model beyond profit - Human Centre design - Theory of Change - Business Model Canvas - Creating a team | **Taking your Social Enterprise to Market** - Developing your marketing strategy - Pitching your idea - Connecting and Networking - Legal Structure and Funding Sources for Social Enterprise - Becoming Impact Investment Ready - Contributors and Additional sources |
| **Course 3** **Social Enterprise: Growing a Sustainable Business** | **Strategies for Sustainability** - Sustaining and Scaling up Strategies - Choosing the right model to scale - Balancing value and growth strategies - Understanding Social Value | **The Challenge of Changemaking** - Bank of Challenges - Partnership Opportunities - Change Management Strategies - Evaluating your Social Enterprise - Measuring Performance | **Achieving Social Impact at Scale** - Commercial, Social and Emotional Marketing - Your Digital Marketing Toolkit - The Impact Investing Framework - Enabling organisations for social enterprise - Social Enterprise business ideas - Contributors and Additional resources |

Source: based on Calvo et al. (2018).

Despite the fact that the courses were free, learners have the opportunity to buy a certificate of achievement for a small fee once the course or program is complete. The courses inserted features facilitating social communication, increasing interactions between all students by enhancing cross-cultural communication, and adopting a twitter-like following system to help track and sustain interactive communication [4].

*3.2. Data Collection and Analysis*

The research methodology adopted for data collection comprised five phases taking place over four years. Firstly, in order to gain the necessary contextual understanding of the learners' experiences on the social enterprise MOOC program and the impact on organisations and individuals, several methods were adopted. An analysis of the existing attendee data of the very first cohort was used to assess the scale and location of the learners. Analysis of the first cohort of 'active learners' showed that 170 countries were covered with 50% from outside of Europe. There were a total of 1364 active learners on the first course, 1005 active learners on the second course, and 995 learners on the third course. Analysis of the early stages found that 15% were active learners engaging in online exercises, debates and conversations. This percentage is comparable with other MOOCs where many learners who stay on the course will watch the content but not always participate in discussions. The evaluation of any MOOC is particularly difficult due to the high drop-out rate for free courses.

Secondly, an analysis of feedback was done looking at a satisfaction survey of the participants conducted in 2017 immediately after completing the course. A detailed analysis of the first cohort drew on 229 responses out of 1364 active learners on the first course, 276 out of 1005 active learners on the second course, and 292 out of 995 active learners on the third course). The results indicate that learners were generally happy with the experience of the courses and would recommend it (87% on the first course, 95% on the second course, and 100% on the third course). In total, 25% of respondents claimed the course was being taken to improve their career and 16% to do their current job better. Moreover, there were some preferences for certain activities of the MOOC. The most useful activities were discussions with peers (32%), video case studies (21%), and articles (18%). The least useful activities were peer-review feedback (14%), toolkits (12%), and quizzes (3%).

Thirdly, semi-structured qualitative interviews with learners were conducted in the year following the first cohort, with a purposeful sample of 10 learners selected to represent different countries and gender. These interviews aimed to explore the short-term impacts in individuals and organisations. The evidence reveals an increase in motivation and confidence among learners, creating positive and psychological effects in them. Another feature that emerged was the notion of increasing their existing knowledge, sharing information with other learners, and contributing to their project. The results reveal

the benefits and opportunities for organisations, particularly aspects that are related to the improvement of their services, efficiency in tasks performed, and acquired financial and human resources.

Fourthly, in order to provide insights into the effectiveness of the Social Enterprise MOOC program, it was necessary to explore the higher level impact (long-term impact) of the online courses. An online impact survey was sent to all learners who completed the courses and gave consent for further contacts. This survey allowed the learners to reflect on their actions after one to three years of the completion of the courses. Analysis is based on the 181 learners responding in full out of the 7500 active learners (see Table 2 for details of the survey sample). The survey was implemented in English via Google Form between May and June 2019 (after three years of the launching of the program). Questions were focused on Levels 3 and 4 from the Freeth/Kirkpatrick Model in order to evaluate the changes in individual behavior and organisations. The survey that utilized a mixture of Likert scales (1 = strongly agree and 5 = strongly disagree) consisted of fourteen multiple-choice questions. Open questions were included in the survey where learners could write about their overall perceptions of the higher level (long-term) impact of the MOOC. The study also examined the extent of the additionality of the MOOC on their practice, compared to other forms of support they have received (i.e. would they have done the changes anyway, has the course helped them make bigger changes, or helped make changes faster).

Quantitative analysis involved descriptive statistics of the online survey results. There were slightly more men within the sample (56%) and in terms of age, the majority of learners that participated on the survey were between 35-55 years old. Regarding the year that the Social Enterprise MOOC program was completed, the vast majority of the respondents (54.1%) responded that it was in 2017.

**Table 2.** Characteristics of the 2019 survey sample.

| Gender | N | Percentage |
|---|---|---|
| Male | 100 | 55.9 |
| Female | 82 | 44.1 |
| **Age** | | |
| 15–35 | 57 | 32.6 |
| 35–55 | 88 | 48.9 |
| 55 | 36 | 25.9 |
| **Completion of the Courses** | | |
| 2017 | 98 | 54.1 |
| 2018 | 64 | 35.4 |
| 2019 | 19 | 10.5 |

Fifthly, qualitative interviews were conducted with a further 10 people. The purpose of these interviews was to evaluate in depth the long-term impact of the MOOC (Levels 3 and 4). Questions were related to the impact the MOOC has had on their social enterprises (start up or existing) and/or the contribution they are making to other SEs because of the course. The interviews were conducted via Skype or telephone. As seen in Table 3, the participants were six females and four males, with ages ranging from 26 to 57. The interviews that lasted between 40 and 70 minutes were digitally recorded and transcribed to ensure accuracy.

The analysis was made by using thematic analysis as an analytical tool to identify key findings from the data collected. Thematic analysis is adequate for this study as it enabled the researchers to capture patterns of meaning across different data sets to answer research questions and assists in understanding people's perceptions, feelings values and experiences. The researchers broke down the interviews into a series of codes. The authors became familiar with the data sets, and then collectively interrogated them for patterns. Then, all the researchers identified patterns and themes in common within and across data sets. Finally, the researchers coded the data, assessed and agreed on a final set of themes and findings.

**Table 3.** Profile of learners interviewed.

| Case No | Gender | Nationality | Job Position | Age | Duration of Interviews |
|---|---|---|---|---|---|
| Case 1 | Male | British | Owner/ Manager | 44 | 45 |
| Case 2 | Female | French | Owner/ Manager | 32 | 60 |
| Case 3 | Female | Ugandan | Owner/ Manager | 26 | 45 |
| Case 4 | Female | Philippines | Owner/ Manager | 46 | 50 |
| Case 5 | Female | Japanese | Owner/ Manager | 40 | 60 |
| Case 6 | Female | United Arab Emirates | Owner/ Manager | 48 | 40 |
| Case 7 | Male | Indian | Manager | 37 | 45 |
| Case 8 | Female | French | Owner/ Manager | 45 | 45 |
| Case 9 | Male | Tanzanian | Owner/ Manager | 57 | 50 |
| Case 10 | Male | United States | Manager | 45 | 70 |

## 4. Presentation of Findings

In order to understand the higher level impact of the Social Enterprise MOOC, the authors used a range of impact indicators: (1) starting new enterprises with awareness of social enterprise, (2) growing an enterprise that includes increasing turnover, new clients, improved services, and increasing the number of beneficiaries (3) doing their job better, and (4) sharing knowledge of social enterprise with others. These indicators were used taking into consideration previous studies related to the impact of entrepreneurship education (See Nabi et al., 2017 [34]).

### 4.1. Starting up Social Enterprises

The courses were initially designed to encourage more people to start up social enterprises. It not only introduced the concept, but it also guided those on the course on the process of setting up the business. When asked whether the courses have helped them to start an enterprise, out of the 181 respondents that participated on the survey, 34% agree, 37% neither agree nor disagree, and 29% disagree. The survey data analysis also shows that out of the 34% claiming the courses helped them start a social enterprise, 76% of these start-ups would not have started without the MOOC. These data are derived from those answering the survey and therefore being the more active learners. Estimating this across all runs of the SE MOOC program with 7500 active learners out of 50,000 registered, there is evidence that at least 26% of active learners or 1945 social enterprises could attribute their ability to start up a social enterprise to the MOOC.

As some learners pointed out:

*We have launched the Prakarsha Foundation. Parallel to it I have also developed an employment and waste management firm.*

*The courses were instrumental in leading the design of the Sustainable Social Protection and Livelihoods Project that is now supporting orphans, vulnerable children, widows and care givers in Suba and Ndhiwa in Kenya.*

*I have been able to apply my knowledge that I developed through the course and has been successfully able to start new movements and I am in design phase to replicate in different countries.*

Not only have new businesses been set up, but there is also evidence that the course has influenced some conventional businesses to shift to social enterprise business models that prioritise social and environmental positive change:

*Before the course, I didn't even understand what social enterprise means. This course gave me a clear understanding on it. Now, I see ways to do my ecological impact in my business.*

The development of business models is a key entrepreneurial process supported by the MOOC. Those learners who started up their social enterprises reported that the courses helped them develop their ideas. As some leaners pointed out:

*The courses have helped me in the way that they have made me realised that I was in the right track and sector and that I had to move to my dreams.*

*This has helped me identify my business model and therefore move from an idea into a project . . . . . .
At the beginning, I did not know what I was doing, but the courses motivated me to continue with the idea and develop it in a social enterprise company.*

A focus of the course is the process of building capabilities to balance the commercial and sustainability objectives. The qualitative data show how the course helped one social enterprise start as they could develop an understanding of social enterprise as well as a deeper understanding of hybrid organising.

*The courses gave me exposure to SE from different countries that helped me to expand my horizon.
It raised my understanding of the balancing act between sustainable impact and the pure profit motive of normal business. The case studies were eye opening on what is achievable with social enterprise.*

The self-assessment elements of the course also assisted them with analyzing possibilities and their own capabilities. Social enterprises are very vulnerable in the early stages and some reported that the courses gave them more confidence which dissuaded them from giving up. For example, one of the founders of an organization based in Dubai and working with Fairtrade projects reported: "*Since I completed the courses I've decided to continue with the project. I was a bit discouraged with my project and didn't know if I should continue with that but after the courses I gained the confidence again to pursue my business idea*".

Furthermore, a greater understanding of social enterprise can have an impact by saving potential entrepreneurs from rushing into starting an unviable business. This, in itself, is a positive social impact, as these individuals have avoided investing time and money in enterprises where they may have been at risk.

*The course made me realise that I wasn't ready to take on something like that.*

*The project did not grow. It was not sustainable. The course helped me to recognise this.*

*4.2. Growing Social Enterprises*

Some of the learners were in leadership roles in social enterprises and other civil society organisations. The proportion of learners in leadership roles is not known, but, in relation to the question whether the course has helped them grow social enterprises, 43% of all survey respondents agree, 38% neither agree nor disagree, and 19% disagree.

*This course has helped me elevate my enterprise to a higher level.*

> *The course has helped me run my already established business better.*

For those respondents who have an existing organization, several sub-themes were identified that included the development of new business models, innovations with new products and services, increases in sales and customers, networking opportunities, and the international expansion of their companies.

> *It has helped me to increase sales. Also, I have been able to connect with other people who are in sustainable fashion.*

> *I have grown since I completed the courses in terms of connections and networking opportunities. I ended up working with a university in my city to promote Fairtrade. That would not have been possible without the completion of the courses that connect me with that university.*

> *This program has helped me grow significantly. Since them, I have been able to expand internationally. I received support from several learners in terms of marketing and strategy and this has been very positive. Growth in terms of sales, clients, etc.*

An important theme that arose from the qualitative data was the self-assessment approach adopted by the participants. While growth was considered to be an important variable for the performance of organisations, self-assessment methods were seen as crucial for the development and scaling up of organisations. Participants were actively involved in analyzing the possibilities and capabilities they had to develop or scale up. As some respondents commented:

> *I have to confess that since I have completed the course I have been able to identify my weakness and take into account what needed to be done to improve the company.*

> *These courses have helped me develop not only professionally speaking but in terms of personal development. I have been able to analyse what needed to be done to improve my company.*

### 4.3. Social Intrapreneurship within Organisations

Many of the learners were not starting or running social enterprises but used the MOOC to make changes within their current employment in private, public, and civil society sectors. These individuals can be referred to as social intrapreneurs, having a positive impact within existing established organisations. When considering whether the course has helped them to do their job better, 66% of learners agree, 28% neither agree nor disagree, and 6% disagree. However, it should be noted that many of those agreeing may also be leading existing or start up social enterprises, and hence may not meet the definition of intrapreneurs. When considering whether the course has helped them to implement bigger changes than they would have done otherwise, 51% of learners agree, 34% neither agree nor disagree, and 13% disagree. When asked whether participants think that they could not have done the changes in the company without the SE course, 49% answered that they think it is because of the course, 37% responded neutral, and 12% disagree. Moreover, and in relation to the question of whether the course has helped them to make changes sooner that they would have otherwise, 57%% of learners agree, 30% neither agree nor disagree, and 13% disagree. Some of the comments from the learners are as follows:

> *I use the knowledge I gained in the course to better serve my clients.*

> *With my expanded knowledge base I have been able to take better and more informed decisions to do my job better. One of my successful consulting assignments in 2017 was shaped by the courses.*

> *It helped me to be more focused, organised and realistic with the project I was involved in.*

The MOOC therefore acts as a way of encouraging innovation within organisations, helping with the development of new services and business models and the development of new partnerships.

*4.4. Sharing Learning*

In relation to the question of whether learners have shared knowledge about SE with others, 79% of learners agree, 12% neither agree nor disagree, and 9% disagree. People reported that the course made them part of a bigger community that facilitated strong and productive relationships.

The course itself encultured a sharing ethos with the exchange of ideas. In some cases, these online relationships were developed with exchange visits.

> *In fact, this course connected me with a social entrepreneur in Japan where I had the opportunity to work for few months and learn from her business.*

> *Yes! It has empowered me and assisted me to believe that anything is possible with hard work, dedication and ingenuity. They have introduced me with key people. These connections have proved priceless exposure internationally. I have been awarded with an international scholarship programme with the School For Social Entrepreneurs and that has been the result of the course.*

> *Learners have transferred knowledge and skills to others, taking on mentoring roles:*

> *I have even passed my knowledge to a group of other 10 social entrepreneurs, and they have used it as well.*

> *I have learned a lot with this course, even to the extent that I am currently advising several women to start up their social enterprises.*

> *My initial dream was only to assist in the development of small and medium size ventures working as a resource person in Nigeria, the course however widened my worldwide that I now focus more on the social relevance/impact of business ideas and not just on typical concerns of every capitalist. I have since been guiding aspiring and emerging entrepreneurs in new light.*

The survey identified four cases of consultants developing new services that are aimed at supporting social enterprise directly, with one working in deprived communities in Nigeria, another working with refugees in Kenya, and one working in Ethiopia.

> *Due to the training, I contribute regularly and play lead roles in Social Protection initiatives and discourses of UN.*

> *I shared the learning with people I work with through our internal social media site. Other people have been able to use the learning I took away from the course in order to provide advice and guidance to voluntary and community groups.*

The MOOC has had an impact on research for sustainable development. Two respondents reported how the course had helped them with their university research and teaching and played a role in setting a research agenda.

> *The course has helped spark discussions with colleagues leading to new projects, new research etc.*

Others shared the ideas from the course when trying to develop collaborations between universities and communities hit by economic crises.

> *I'm trying to convince some friends, the mayors of little villages and some universities to work together to find new solutions to the actual economic crises in South Italy. I talked with them about some ideas I found during this course and I recommended to take a look to MOOCs like this.*

The MOOC has also shaped the funding decisions of the British Council, who provided initial support, and shaped the future work of the people involved in developing and delivering the course.

## 5. Discussion

This paper has examined the impact of digital education by examining the case of a MOOC on social enterprise. The design of the course brought together forms of learning from filmed case studies, with wider research on social enterprise. This was combined with online interaction elements around sharing experiences and working on practical exercises. This combination demonstrates a shift from instructor-led online lectures to the development of MOOCs that are learner centered [20]. This, therefore, brings together the different elements of entrepreneurship education, as set out by Nabi et al. [31] in their review of different models. Experiential pedagogies have been reported to have the potential to create a bigger impact at higher levels because students can focus on solving real-life entrepreneurial problems and situations.

To assess the impact of the case study MOOC, this paper draws on the Freeth/Kirkpatrick model (FKM) that identifies the need to look at the changes in behaviour and the results in terms of organizational performance [33]. This requires a longitudinal approach, assessing changes over time.

In terms of the first research question, (how can a MOOC encourage social enterprises to start up?), this study shows how a MOOC does lead to new start-ups with an estimation of 1945 social enterprises attributed to this course, taking a conservative estimate. The MOOC was reported to provide confidence for those starting enterprises, which is a crucial element of the start-up process. However, evaluations of all MOOCs are challenging as learners may dip in and out of the course and may not be willing to be contacted by evaluators. The findings also show that some learners have used the course to shift an existing business towards becoming more focused on social enterprise activity. This supports findings from around the world showing how social entrepreneurs may be using private sector legal forms where no specific social enterprise legal forms exist, but shifting their mission to be focused on sustainability [12].

The second research question asked 'how does a MOOC contribute to the impact stakeholders and learners had on their existing projects and organisations?' Some of the learners were already running social enterprises, and there is evidence that these existing social entrepreneurs attribute the growth of their enterprises to what they learnt on the course. The qualitative data show that the MOOC has supported growth in terms of increased sales, increased number of users/beneficiaries, increased geographic scale (including international expansion) and increased scope through diversification. While this supports previous findings related to social enterprise growth [14,15], the findings also show that there is a bigger group of learners who are using the MOOC within their current private, public, or civil society organisations. These learners are social intrapreneurs, bringing social enterprising ideas into these organisations.

The third research question asked 'how have learners used the MOOC for sharing knowledge on sustainable development and social enterprise?' A surprising finding was the extent to which the learners had been sharing the learning with a wide range of other organisations and individuals. For some, this was through informal relationships, while for others it was more formalized mentoring or part of their job to support other social enterprises. This demonstrates a way of scaling up social entrepreneurial ideas and principles that would never be possible with conventional courses.

## 6. Conclusions

This paper set out to examine the role of digital education on social enterprise and sustainability by examining a case study of a MOOC. Lessons can be learnt about the nature of this form of support and the ways that it can complement other forms of support. Firstly, the design of a course on social enterprise needs to tap into the ways entrepreneurs learn. This involves learning from other social entrepreneurs, having practical examples, and being able to discuss issues with others. The online fora of the Social Enterprise MOOC provided this environment, with evidence that online discussion led to longer term relationships and in one case invitations to visit a co-learner in another country.

The assessment of the impact of the course shows the range of ways that courses can help grow the social enterprise movement. Sustainable development requires the growth of pro-environmental

and pro-social activities that can address grand challenges, and this paper demonstrates how social and sustainable entrepreneurship can be scaled up. This may be through individual organisations, as well as scaling through other organisations. Scaling is also possible through sharing ideas outside the organisation. This is a core distinction from the private sector where the competitive environment can hinder the sharing of knowledge.

This study has limitations in terms of accessing learners and having a bigger sample of learners responding to the interviews and survey several years after they have finished the course. Moreover, there are some concerns regarding the low number of active learners in comparison with the number of people that registered on the courses (7500 out of 50,000 people). Another aspect that was not taken into account was the personal behaviour and livelihoods of learners that came from different contexts. New MOOCs could have a greater emphasis on collecting contact details of learners and ensuring that they give consent for future surveys. Future research may also be able to look at the wider benefits of sustainability and social enterprise related MOOCs on the personal behaviour and livelihoods of those learners. Further research is also needed on how the learning from courses is cascaded down to others. The vital ingredient of MOOCs is their ability to be shared broadly and openly, with those leading the course allowing learners to take the content and ideas of the course and use as they feel fit. By opening up ideas and allowing MOOCs to go viral, educators have the chance to shape sustainability on a massive scale.

**Author Contributions:** Conceptualization, F.L. and S.C.; formal analysis, S.C. and F.L.; investigation, S.C. and A.M.; methodology, S.C., A.M. and J.W.; supervision, A.M., F.L. and J.W.; validation, F.L, A.M. and J.W.; writing—original draft, S.C., A.M. and F.L.; and writing—review and editing, S.C, A.M., F.L. and J.W. All authors have read and agreed to the published version of the manuscript.

**Funding:** This work was supported by the British Council [grant number 56734], the ESRC Centre for the Understanding of Sustainable Prosperity (CUSP) and Middlesex University Research Facilitation Fund.

**Acknowledgments:** We want to thank the British Council India and FutureLearn for their support in developing the MOOC and all participants for their generous assistance and substantial amount of time they invested into study participation.

**Conflicts of Interest:** The authors declare no conflict of interest.

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
