# Peer review of "Educating at Scale for Sustainable Development and Social Enterprise Growth: The Impact of Online Learning and a Massive Open Online Course (MOOC)"

_sustainability, doi:10.3390/su12083247_

Round 1
Reviewer 1 Report
The paper is very interesting and covers an important topic.
My general opinion about the paper is very positive.
I have however two concerns:
(i) the sample considered is not large enough to allow robust conclusions;
(ii) the interpretation of the results could be imporved namely through a better link with previous works on the same topic.
While the first aspect is a general shortcoming of the paper that is hard to overcome at the current moment, some additional effort could be done regarding my second comment.
Author Response
We would like to thank the editor and reviewers for their constructive comments on our paper and their very useful suggestions for revisions.
We have revised the paper to address fully these outstanding concerns. Specifically, we have responded to the key issues raised concerning: (1) the methodology used; and (2) the explanation of findings.
Full details of our response to each of the comments made are set out below and all changes made are highlighted in the text.
If you require any further information, please do not hesitate to ask.
Yours sincerely
Sara Calvo
|
Referees comments
|
Response |
|
The interpretation of the results could be improved namely through a better link with previous works on the same topic.
|
We have strengthened the analysis and discussion, referring back to the literature and identifying the contribution of the paper. This includes a discussion of the shift from instructor led to learner led approaches, the need to look at long term impacts of sustainability and entrepreneurship education, and Social enterprise growth theory. |
|
The methods should be explained. As it is, I am not convinced that the percent of participant, survey completers, and reliability of the data presented.
|
Further details are given and caveats provided. These are shown to relate to common challenges facing the evaluation of MOOCs. |
|
Question 4 is not noted.
|
This question has been deleted. Only three questions have been raised in the paper. |
|
Sentence structure in many places is a bit awkward and confusing. Proofreading is needed.
|
The paper has been edited carefully and corrections made throughout |
|
In terms of whether or not the courses helped those working in leadership roles in organisations only 43% agreed the courses were helpful This is not a strong finding in reality. Something should be noted as to this effect.
|
The paper now explains that the findings relate to 43% of ALL respondents, many of whom are not in leadership positions and so would have to opt for the option of ‘disagree’. |
|
Add an introductory comment about the criticisms of MOOCs (very high drop out rate). This MOOC is student centred is encourage as some other such as UDEMY or LAUNCHX are not.
|
Changes have been done. See the document.
|
|
Explain why only 10 people were interviewed in the first time and 6 students the second group. |
The paper now explains that qualitative work was exploratory and with a small number of interviews carried out in depth. |
|
Also, of the 50,000 people registered only 7,500 active learners. This should be explained.
|
The paper explains that all MOOCS face high numbers of people who use the course but in the more intense participatory way. |
|
A little more space could be devoted to the limitations of the study (even though the authors make reference to it).
|
This has been added on the paper. |
|
The notion of sustainability, just like the notion environmental awareness, raises issues/problems, and hence limitations, as regards the development of the concept per se, and these need to be considered in the context of education. In general. (see Hadzigeorgiou, Y. & Skoumios, M. (2013). The development of environmental awareness through school science: Problems and possibilities. International Journal of Environmental and Science Education 8(3), 405-426.)
|
This has been added on the paper. |
|
2) A table, following table 1 (teaching model), that could provide info about the content of the course per se. Content does play a crucial role (even though the purpose of the article was to examine the role of digital education and not the role of the content). A table 2 following the teaching model could very useful to the general reader.
|
This table has been modified including detail information about the content of the courses and text has been added explaining the information that was in Table 1. |

Reviewer 2 Report
This is an interesting artice about the impact of digital education on SE through an examination of the rle of MOOC. I believe the article can be published, and my ony recommendations/suggestions are:
1) A little more space could be devoted to the limitations of the study (even though the aothors make reference to it).
(MOREOVER, The notion of sustainability, just like the notion environmental awarenss, raises issues/problems, and hence limitations, as regards the deveopment of the concept per se, and these need to be considered in the context of education, in general. (see Hadzigeorgiou, Y. & Skoumios, M. (2013). The development of environmental awareness through school science: Problems and possibilities. International Journal of Environmental and Science Education 8(3), 405-426.)
2) A table, following table 1 (teaching model), that could provide info about the content of the course per se. Content does paly a crucila role (even though the purpose of the article was to examine the role of digital education and not the role of the content). A table 2 following the teachng model could very useful to the general reader.
Thank u for giving me the opportunity to review your work.
Author Response

(The authors gave the same response as above.)

Reviewer 3 Report
MOOC research is needed and can make a significant contribution to the field along with Sustainability and development of social enterprises. If the methods can be adequately explained it will be a good paper. As it is, I am not convinced that the percents of participant, survey completers, and reliability of the data presented. That is why I stated that I have ethical concerns. If the concerns outlined in the paper are addressed, it will be a significant contribution to the field.

Author Response

(The authors gave the same response as above.)

Reviewer 4 Report
On Line 56, page 2, you note that you seek to answer 4 questions but only 3 questions are noted. So you should either add the 4th question or reduce the 4 to 3.
Sentence structure in many places is a bit awkward and confusing. It would help if you either proof the paper very carefully yourself or ask someone who is more objective to do so - especially someone with strong grammar skills in English. Some words also are not accurate, appropriate or helpful in providing meaning to the paper. Another example is on Page 2, Line 56 where the author says that they sought to answer 4 reflective questions and then states only 3, not 4. So again, rigorous proofing would help.
Third. in terms of whether or not the courses helped those working in leadership roles in organizations only showed 43.1% agreeing that the courses were helpful. So this is not a strong finding in reality. Something should be noted as to this effect.
Fourth, some introductory comment about the criticisms of MOOCs should be noted in the introduction. MOOCs are known for drawing extremely large numbers of participants but also experience a very high drop out rate. So they have been a mixed bag. This will serve to highlight the better than normal rates for sustainability MOOCs. In addition, many of the MOOCs that many are familiar with from LaunchX, Udemy for example, are not as student-centered as this paper suggests. To hear that the MOOCs that they reviewed in sustainability are more interactive and are closer to a "flipped classroom" experience is highly encouraging.
The average reader will look at the forms utilized to collect data and wonder why only 10 people were interviewed the first time and 6 students the second group of 10 (since 4 of the 10 were connected to the courses in some way). Also notations as to why you could not get a higher response rate for the surveys knowing that 50,000 registered for the MOOCs and you estimated that there were probably 7500 active learners would be helpful. That will help mitigate the concern that such a reader might have in looking at the numbers.
Author Response

(The authors gave the same response as above.)

Round 2
Reviewer 3 Report
I still do not know what percent of the survey sent out were returned. That number is important. I still do not know in the results section whether the answers are contingent on the 1st survey, the 1st interviews, the 2nd survey given online or the 2nd set of interviews. This is still confusing. I need to know the number of participant when percents are given. They are mostly not available. Table with percents should include the N (number of participants)

Author Response
We would like to thank the editor and the reviewers for their minor comments.
We have revised the paper to address fully concerns.
All changes made are highlighted in red in the text following the indications from the reviewers with regards to the number of people that participated in the surveys.
If you require any further information, please do not hesitate to ask.
Yours sincerely
Sara Calvo